# Motus Vita Est: Fruit Flies Need to Be More Active and Sleep Less to Adapt to Either a Longer or Harder Life

**Lyudmila P. Zakharenko [1], Dmitrii V. Petrovskii [1], Margarita A. Bobrovskikh [1], Nataly E. Gruntenko [1], Ekaterina Y. Yakovleva [2], Alexander V. Markov [2,3] and Arcady A. Putilov [4,5,*]**

[1] Department of Insect Genetics, Institute of Cytology and Genetics of the Siberian Branch, The Russian Academy of Sciences, Novosibirsk 630000, Russia

[2] Department of Biological Evolution, The Moscow State University, Moscow 101000, Russia

[3] Borisyak Paleontological Institute of the Russian Academy of Sciences, Moscow 101000, Russia

[4] Research Group for Math-Modeling of Biomedical Systems, Research Institute for Molecular Biology and Biophysics of the Federal Research Centre for Fundamental and Translational Medicine, Novosibirsk 630000, Russia

[5] Laboratory of Sleep/Wake Neurobiology, Institute of Higher Nervous Activity and Neurophysiology of the Russian Academy of Sciences, Moscow 101000, Russia

\* Correspondence: putilov@ngs.ru; Tel.: +49-30-53674643 or +49-30-61290031

**Abstract:** Background: Activity plays a very important role in keeping bodies strong and healthy, slowing senescence, and decreasing morbidity and mortality. *Drosophila* models of evolution under various selective pressures can be used to examine whether increased activity and decreased sleep duration are associated with the adaptation of this nonhuman species to longer or harder lives. Methods: For several years, descendants of wild flies were reared in a laboratory without and with selection pressure. To maintain the "salt" and "starch" strains, flies from the wild population (called "control") were reared on two adverse food substrates. The "long-lived" strain was maintained through artificial selection for late reproduction. The 24 h patterns of locomotor activity and sleep in flies from the selected and unselected strains (902 flies in total) were studied in constant darkness for at least, 5 days. Results: Compared to the control flies, flies from the selected strains demonstrated enhanced locomotor activity and reduced sleep duration. The most profound increase in locomotor activity was observed in flies from the starch (short-lived) strain. Additionally, the selection changed the 24 h patterns of locomotor activity and sleep. For instance, the morning and evening peaks of locomotor activity were advanced and delayed, respectively, in flies from the long-lived strain. Conclusion: Flies become more active and sleep less in response to various selection pressures. These beneficial changes in trait values might be relevant to trade-offs among fitness-related traits, such as body weight, fecundity, and longevity.

**Keywords:** fruit fly model; locomotor activity; sleep duration; artificial selection; longevity; adverse diet

## 1. Introduction

From an evolutionary perspective, an organism is designed to extract energy from the environment and use it to produce offspring. In order to increase reproductive success, it is necessary to maintain the balance between energy intake and expenditure, and to optimally allocate energy across the life span whilst growing up to take care of the body, reproduce, and make it easier for the offspring to reproduce [1,2].

Species use very diverse strategies to allocate energy to the essential tasks of growth, maintenance, movement, reproduction, and care of offspring [3]. Natural selection often does not favor the reduction in energy expenditure. The tendency to increase energy expenditure can be exemplified by the evolution of physical activity in its most common form, locomotor activity, which represents an important component of daily energy expenditure in animals and humans [4,5]. Organisms that have evolved with increased levels

of locomotor activity would be expected to cope with changes in their environment better than organisms with reduced levels of this activity [4,6]. The increase in locomotor activity has often played a major role in macroevolution of broad taxonomic groups of animals. For example, selection for high levels of aerobically supported locomotor activity can be a key factor causing the evolution of vertebrate endothermy [7,8].

In humans, many pathologies, such as Parkinson's disease, Huntington's disease, activity disorders, and depression, are associated with the deficits of locomotion [9,10]. Therefore, understanding the genetic and environmental contributors to locomotor behavior is important from the perspectives of evolutionary biology and human health [11–14]. Habitual levels of locomotor activity and health-related physical fitness traits appear to be genetically heritable [15–18]. In human studies, such traits have been associated with various health benefits, while, in conditions of food abundance and limited physical activity, the excessive amounts of fat accumulate and can even cause obesity, mostly due to energy imbalance [19–23].

A low metabolism underlies a slow pace of growth, reproduction, and aging in humans. However, their evolutionary history has deemed that they be more active and sleep less compared to their close relatives, apes [24]. Although higher physical activity requires more energy, humans have not evolved to prevent senescence by its reduction. Instead, they have evolved to remain physically active for the improvement of their health and extension of their lifespans [24–26]. Consequently, this integral component of most behaviors plays a very important role in keeping the body strong and healthy, slowing senescence, and decreasing morbidity and mortality. Humans in and after middle age require physical activity to increase their lifespans and reduce the risk of disease and death [27–30].

In order to manage peoples' health today, it is necessary to understand (1) why our and many other species evolved to be physically more active and, therefore, to sleep less and (2) how and why the evolutionary changes in longevity and health were accompanied by changes in activity and sleep patterns.

Selection for a single trait can often result in correlated changes in other traits [31,32]. The idea that several important traits can evolve together in response to selection is not controversial, but it is difficult to directly test this in human evolutionary studies [33]. Many obvious practical and ethical obstacles limit the scope for experiments using humans in such studies. Therefore, many insights have been derived from experiments on model organisms, and much of what we know about the underlying co-evolution of complex traits had come from studies using these organisms. Laboratory selection has been long used for exploring hypotheses about various adaptations because experimental evolution can quickly and reproducibly shape phenotypes in model species [34,35]. Particularly, some animal species offer unique models for the experimental research of how locomotor activity has co-adapted with other complex traits, such as longevity, fecundity, energy conservation, body weight, resistance to stress and infections, etc. [36,37].

*Drosophila melanogaster* (fruit fly or, more correctly, vinegar fly) is a human commensal of eastern sub-Saharan African origin [38,39]. For more than a century, it has been used as a model organism to study a diverse range of biological processes, including evolution through natural and artificial selection, genetics and inheritance, locomotion and other behaviors, learning and cognitive skills, development and aging, disease vulnerability, etc. Due to its simple and rapid life cycle, cosmopolitan distribution, ease of maintenance in the laboratory, and well-understood evolutionary genetics, *Drosophila melanogaster* has become one of the most powerful animal models for studying the evolution of complex traits [36,40]. Humans and fruit flies may not look very similar. However, it is well-established that most of the fundamental biological mechanisms and pathways that control the behavior, development, survival, and reproduction of an organism are conserved between these two species across their evolution [41]. Therefore, the genes affecting locomotion and aging in *Drosophila* often have human orthologs and will elucidate corresponding mechanisms in humans [35,36,42–44].

*Drosophila melanogaster* exhibits a strong response to artificial selection (i.e., selective breeding) for high and low levels of locomotor reactivity [41]. Genes affecting locomotor activity are also likely to be involved in many other forms of behavior, neurogenesis, metabolism, development, major cellular processes, etc. [42]. In general, such artificial selection experiments can be used to mimic evolutionary processes to test hypotheses about the correlated evolution of complex traits [45–48]. Since this experimental approach can help to reveal the underlying genetic correlations between a trait under selection and other traits, *Drosophila melanogaster* became a favorite model organism for experimental selection in studies of complex traits associated with the aging process [49–55]. For the elaboration of the mechanisms underlying aging and senescence, artificial selection was employed for *Drosophila* to create extended longevity strains [56–59]. It was, in particular, demonstrated that reproduction late in life increases longevity [52,56,60–62].

Moreover, the co-evolution of complex traits in *Drosophila melanogaster* can be investigated using another approach, laboratory (controlled) natural selection (or, more accurately, natural selection in a controlled environment). Instead of selecting flies according to their values for a given trait, they are allowed to evolve for many generations under experimentally imposed environmental treatments, such as changes in nutrient and temperature, different light–dark cycles, etc. This approach allows the experimenter to impose carefully controlled selective conditions in the laboratory and then observe evolutionary responses in real time [63]. One example of evolutionary responses in laboratory settings is the investigation of adaptation of selected strains to different diets [64–70]. Nutrition is a primary determinant of reproductive capacity, life span, rate of development and aging. The amount and quality of consumed nutrients have a strong impact on life-history (or "fitness components") traits, such as disease vulnerability, stress resistance, fertility, body weight, reproduction, and longevity [71–77]. Poor nutrition during development generally results in detrimental fitness effects, which include decreased values of traits such as body weight, fecundity, and lifespan [78–80]. Therefore, *Drosophila melanogaster* has been widely used in experiments on controlled natural selection to address many fundamental questions in evolutionary biology, physiology, and ecology [81–83].

Overall, two approaches, artificial selection and controlled natural selection, can be used to test the adaptive changes in various complex traits, including activity, sleep, body weight, fecundity, and longevity.

In the present study, these approaches to modeling the evolutionary process in *Drosophila melanogaster* were applied to examine whether locomotor activity can increase and, consequently, whether sleep duration can decrease in response to artificial selection toward a longer lifespan and in response to controlled natural selection on adverse food substrates. The following major hypothesis was tested:

(1) When fruit flies evolve to have a longer lifespan or to live and reproduce on an adverse food substrate, their locomotor activity increases and, consequently, their sleep duration decreases.

The alternative hypothesis can be also proposed:

(2) Fruit flies are becoming less active and sleep more.

## 2. Results

The characteristics of the 24 h cycles of locomotor activity and sleep in flies from the three selected strains (long-lived, salt, and starch) have diverged from the characteristics of the control flies that were left on the control food substrate without any selection imposed on their development and reproduction. Compared to the control flies, flies from each of the three selected strains demonstrated higher locomotor activity and shorter sleep duration (Tables 1 and 2). The most profound increase in locomotor activity was found in starch flies (Figure 1), which, in fact, were selected for early reproduction due to their reduced lifespan (Figure S1).

**Table 1.** Two-way rANOVAs of locomotor activity and sleep.

| rANOVAs on 48 30 min Intervals of the 24 h Patterns | | | | | | | |
|---|---|---|---|---|---|---|---|
| **Sex** | **Male** | | | | **Female** | | |
| **Measure** | **Activity** | | **Sleep** | | **Activity** | | **Sleep** | |
| **Factor** | **F** | **df** | **F** | **df** | **F** | **df** | **F** | **df** |
| "Time point" | 109.3 *** | 47/13,677 | 207.0 *** | 47/13,677 | 67.5 *** | 47/8366 | 128.4 *** | 47/8366 |
| "Strain" | 9.4 *** | 3/291 | 14.6 *** | 3/291 | 23.9 *** | 3/179 | 18.4 *** | 3/179 |
| Interaction | 14.3 *** | 141/13,677 | 13.3 *** | 141/13,677 | 15.1 *** | 141/8366 | 13.4 *** | 141/8366 |

| **Pairwise comparisons** | | | | Sleep (min per 30 min) | | | |
|---|---|---|---|---|---|---|---|
| Male | Activity | | Control | Long-lived | Salt | Starch | | |
| Strain | Mean | SEM | | | | | Mean | SEM |
| Control | 21.13 | 2.14 | - | 6.07 *** | 2.30 | 3.48 ** | 20.14 | 0.65 |
| Long-lived | 30.64 | 2.20 | −9.51 *** | - | −3.77 *** | −2.59 | 14.07 | 0.67 |
| Salt | 24.38 | 2.14 | −3.25 | 6.26 | - | 1.18 | 17.84 | 0.65 |
| Starch | 36.70 | 2.35 | −15.57 *** | −6.05 | −12.3 *** | - | 16.66 | 0.71 |
| | Activity (per 30 min) | | | | | | Sleep | |

| Female | Activity | | Control | Long-lived | Salt | Starch | Sleep | |
|---|---|---|---|---|---|---|---|---|
| Strain | Mean | SEM | | | | | Mean | SEM |
| Control | 13.60 | 2.65 | - | 5.20 *** | 3.43 ** | 7.16 *** | 22.09 | 0.71 |
| Long-lived | 23.89 | 2.22 | −10.28 ** | - | −1.77 | 1.96 | 16.89 | 0.69 |
| Salt | 17.72 | 2.18 | −4.11 | 6.17 | - | −3.73 ** | 18.67 | 0.70 |
| Starch | 37.96 | 2.18 | −24.35 *** | −14.07 *** | −20.24 *** | - | 14.94 | 0.70 |
| | Activity (per 30 min) | | | | | | Sleep | |

Notes: Upper part: Male and Female: Two two-way rANOVAs (for each sex) with the repeated measure "Time point" (48 30 min intervals). "Strain" is the independent factor (referred to as control, long-lived, salt, and starch; n = 78, 74, 78, and 65 for Male and 44, 47, 46, and 45 for Female, respectively); degrees of freedom (df) were corrected using Greenhouse–Geisser correction controlling for type 1 errors associated with the violation of the sphericity assumption, but the original degrees of freedom are reported in this table; F: F-ratio. Lower part: Mean and SEM: Daily averaged locomotor activity or sleep, per 30 min, and standard error of this mean. Results of post hoc pairwise Bonferroni comparisons of strains on daily averaged locomotor activity and sleep (below and above the diagonal, respectively); ** $p < 0.01$, *** $p < 0.001$ for F or t (these symbols are shown behind F-ratio or behind the estimates of difference between strain-averaged values). Results are illustrated in Figures 1a and 2a.

**Table 2.** Three-way rANOVAs of locomotor activity and sleep.

| rANOVAs on 48 30 min Intervals of the 24 h Patterns | | | | | | | |
|---|---|---|---|---|---|---|---|
| **Adverse Food Substrates** | | | | | **Strains/Hybrids** | | |
| **Measure** | **Activity** | | **Sleep** | | **Activity** | | **Sleep** | |
| **Factor** | **F** | **df** | **F** | **df** | **F** | **df** | **F** | **df** |
| "Time point" | 47.6 *** | 47/7567 | 105.7 *** | 47/7567 | 132.8 *** | 47/11,468 | 281.5 *** | 47/11,468 |
| "Strain" | 11.9 *** | 3/161 | 10.3 *** | 3/161 | 29.7 *** | 3/244 | 32.5 *** | 3/244 |
| Interaction | 13.3 *** | 141/7567 | 15.8 *** | 141/7567 | 15.6 *** | 141/11,468 | 12.1 *** | 141/11,468 |

| **Pairwise comparisons** | | | | Sleep (min per 30 min) | | | |
|---|---|---|---|---|---|---|---|
| Adverse food | Activity | | Control | Long-lived | Salt | Starch | | |
| Strain | Mean | SEM | | | | | Mean | SEM |
| Control | 13.47 | 2.14 | - | 1.77 | 6.94 *** | 4.82 *** | 21.59 | 0.78 |
| Long-lived | 18.16 | 1.87 | −4.52 | - | 5.17 *** | 3.05 ** | 19.70 | 0.68 |
| Salt | 25.90 | 1.90 | −12.74 *** | −8.21 * | - | −2.12 | 14.81 | 0.69 |
| Starch | 28.15 | 1.83 | −14.74 *** | −10.22 ** | −2.01 | - | 16.83 | 0.66 |
| | Activity (per 30 min) | | | | | | Sleep | |

**Table 2.** *Cont.*

| Measure | rANOVAs on 48 30 min Intervals of the 24 h Patterns | | | | | | | |
|---|---|---|---|---|---|---|---|---|
| | **Adverse Food Substrates** | | | | **Strains/Hybrids** | | | |
| | **Activity** | | **Sleep** | | **Activity** | | **Sleep** | |
| **Factor** | **F** | **df** | **F** | **df** | **F** | **df** | **F** | **df** |
| | | | | | **Sleep (min per 30 min)** | | | |
| Strains/hybrids | **Activity** | | **Control** | **Long-lived** | **Hybrids** | **Starch** | **Sleep** | |
| Strain | Mean | SEM | | | | | Mean | SEM |
| Control | 8.90 | 1.54 | - | 7.72 *** | 6.55 *** | 5.78 *** | 23.30 | 0.61 |
| Long-lived | 22.74 | 1.60 | −14.09 *** | - | −1.17 | −1.95 | 15.61 | 0.64 |
| Hybrids | 23.60 | 1.10 | −14.71 *** | −0.62 | - | −0.77 | 16.74 | 0.44 |
| Starch | 28.09 | 1.57 | −19.19 *** | −4.48 | −5.10 | - | 17.52 | 0.63 |
| | | | **Activity (per 30 min)** | | | | **Sleep** | |

Notes: Upper part: Adverse food substrates and strains/hybrids: Two three-way rANOVAs with the repeated measure "Time point" (48 30 min intervals). "Strain" is one of two independent factors (referred to as control, long-lived, salt, and starch, n = 37, 45, 44, and 47, and control, long-lived and starch, and F1 hybrids between long-lived and starch, n = 52, 48 and 50, and 102, respectively); an additional independent factor was either "Diet" (three adverse diets: salt or starch or low protein/high carbohydrate) or "Sex" (male or female on the standard diet). Interaction: Interaction of repeated measure with "Strain"; F: F-ratio. Lower part: Results of post hoc pairwise Bonferroni comparisons of strains on daily averaged locomotor activity and sleep (below and above the diagonal, respectively); * $p < 0.05$, ** $p < 0.01$, *** $p < 0.001$ for F or t (these symbols are shown behind F-ratio or behind the estimates of difference between strain-averaged values, respectively). See also notes for Table 1 and illustrations of rANOVAs results for strains/hybrids in Figures 1b and 2b.

Consequently, three selected strains (long-lived, salt, and starch) have also evolved a reduced sleep duration (Figure 2 and Tables 1 and 2).

At least two of three selected strains showed significant differences from the control strain not only in mean levels but also in the 24 h patterns of locomotor activity and sleep (Tables 1 and 2, and Figures 1 and 2). For instance, the selective breeding of the long-lived flies for late reproduction was associated with changes in the positions of morning and evening peaks of locomotor activity. The long-life-selected flies demonstrated a phase advance of the morning peak combined with a phase delay of the evening peak (Figures 1 and 2). Having a higher overall level of locomotor activity, flies from the starch strain were also more active in the morning and evening hours than the control flies. However, flies from this strain did not show a clear difference from the control flies in the positions of morning and evening peaks (Figures 1 and 2).

Importantly, all such differences between selected and unselected strains in the 24 h patterns showed stability across substrate (Table 2) and sex (Figures 1 and 2). This prompted us to examine the 24 h patterns in the hybrid flies obtained by the bidirectional breeding of flies from the long-lived and starch strains. As expected, these flies demonstrated intermediate 24 h patterns from their parent strains (Table 2 and Figure 1b,c and Figure 2b,c).

The results for the fly weight (Table 3), concentration of two sugars in the fly's hemolymph (Table 3), survival rate (Supplementary Figure S1 and Tables S1) and fecundity (Supplementary Table S2) provided the possibility to relate changes in the locomotor activity of flies from the selected strains (Tables 1 and 2) to the changes in several life-history traits. For instance, we found that an adaptation permitting us to allocate a higher amount of energy in locomotion can be, at least partly, explained by adaptive morphophysiological changes (i.e., a decrease in body weight combined with an increase in concentration of trehalose, the major insect sugar; Table 3).

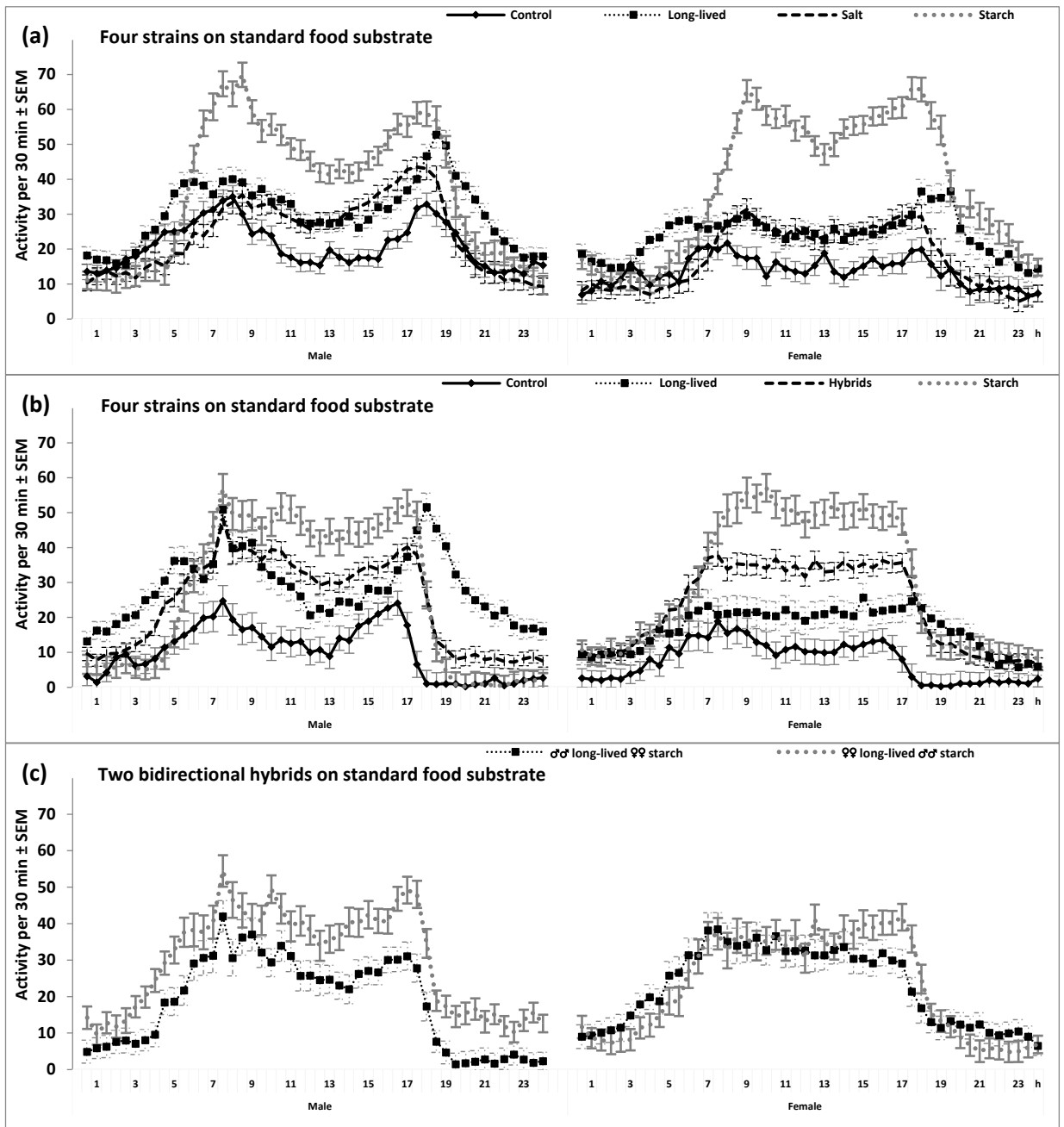

**Figure 1.** The 24 h patterns of locomotor activity. Locomotor activity was defined as the number of beam crossings per 30 min intervals. SEM: Standard error of mean of strain-averaged locomotor activity on 30 min intervals of the 24 h cycle (calculated from data obtained on one min intervals of the 2nd to the 4th day of the recording in constant darkness). (**a**). From the results of two two-way rANOVAs. The repeated measure was "Time point"; "Strain" was the only independent factor. Control, long-lived, salt, and starch strains: Selection was not carried out for the control strain, and three strains were selected for a longer lifespan and to adapt to adverse (salt and starch) food substrate, respectively; n = 78, 74, 78, and 65 for Male and 44, 47, 46, and 45 for Female, respectively. (**b,c**): From the results of three-way rANOVAs with another independent factor "Sex" (Male or Female). (**b**): Control, long-lived and starch strains, and the hybrids of two later strains (n = 52, 48 and 50, and 102, respectively). (**c**): F1 hybrids from bidirectional breeding: parents either ♂♂long-lived ♀♀starch or ♀♀long-lived ♂♂starch (n = either 50 or 52, respectively); h: Clock hour. See also Tables 1 and 2 for the results of two- and three-way rANOVAs illustrated in Figure 2a,b, respectively.

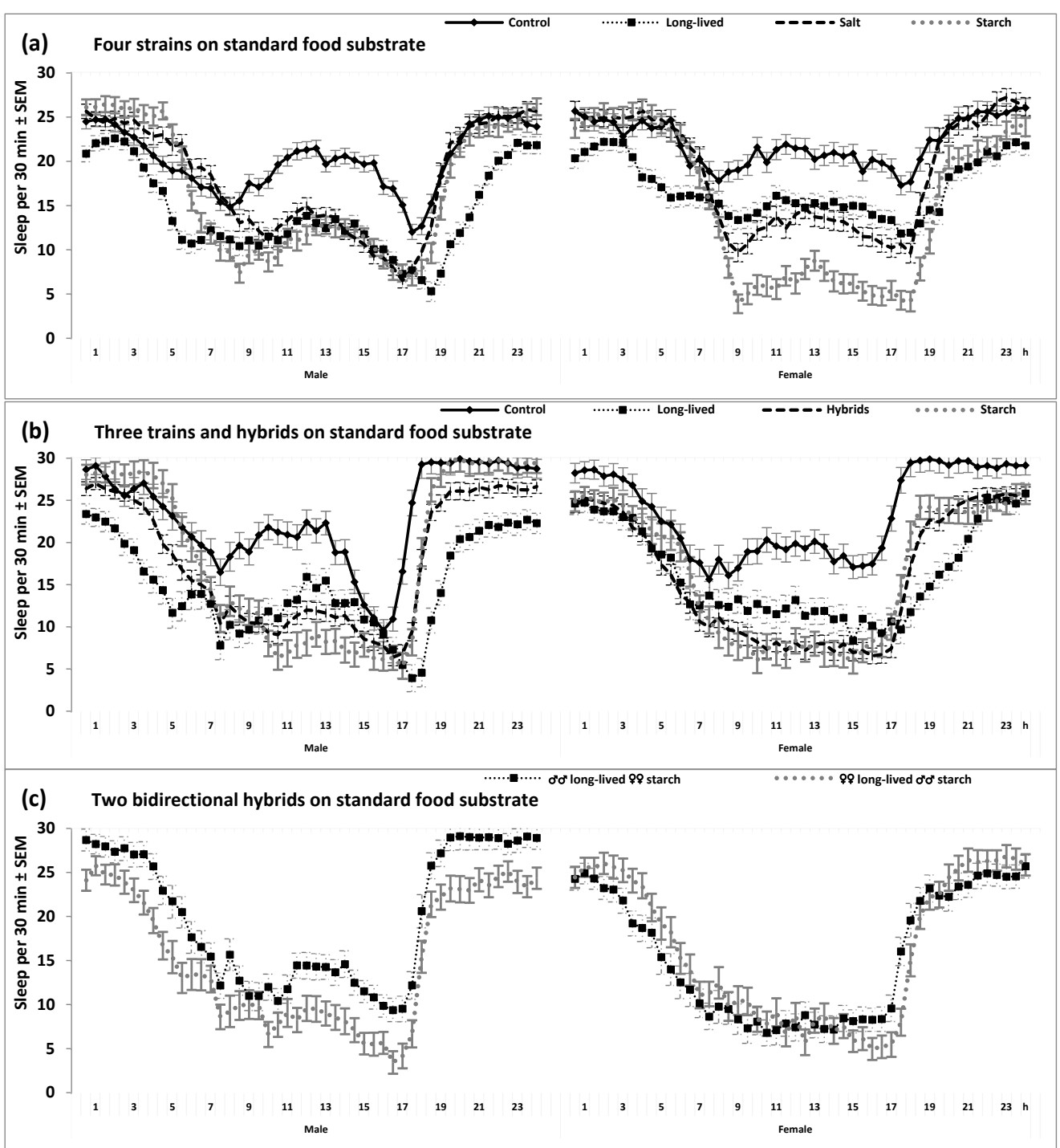

**Figure 2.** The 24 h sleep patterns. (**a**). From the results of two two-way rANOVAs. (**b**,**c**): From the results of three-way rANOVAs with another independent factor "Sex" (Male or Female).

Moreover, the results on survival rate (Table S1 and Figure S1) revealed the expected strain-specific differences between flies from the selected and unselected strains that have persisted over several generations in the Novosibirsk institute. For example, the selection of flies through breeding older adults resulted in an evolutionary increase in the survival of flies from the long-lived strain (i.e., a longer lifespan compared to the control flies). In contrast, the strains adapted to living on adverse food substrates had shorter lifespans than the control strain (Table S1 and Figure S1). The lifespan was found to be the shortest in flies from the starch strain reared on the unhealthiest of all diets (Figure S1).

**Table 3.** Results of three-way ANOVAs of fly weight and two main inset sugars.

| Measurement | Strain | Male Mean | Male SEM | Female Mean | Female SEM | Factor | F | df | p |
|---|---|---|---|---|---|---|---|---|---|
| Fly weight, mg | Control | 0.856 | 0.036 | 1.427 | 0.033 | "Sex" | 322.4 | 1/241 | <0.001 |
| | Long-lived | 0.748 | 0.037 | 1.144 | 0.034 | "Diet" | 13.9 | 1/241 | <0.001 |
| | Salt | 0.741 | 0.033 | 1.132 | 0.035 | "Strain" | 14.9 | 3/241 | <0.001 |
| | Starch | 0.830 | 0.035 | 1.218 | 0.032 | Interaction | 3.4 | 3/241 | 0.02 |
| Glucose, µg/mg fly | Control | 4.40 | 0.24 | 6.24 | 0.17 | "Sex" | 90.0 | 1/103 | <0.001 |
| | Long-lived | 4.14 | 0.23 | 4.95 | 0.19 | "Diet" | 75.3 | 1/103 | <0.001 |
| | Salt | 3.61 | 0.16 | 5.52 | 0.17 | "Strain" | 10.3 | 3/103 | <0.001 |
| | Starch | 4.99 | 0.18 | 5.54 | 0.16 | Interaction | 7.3 | 3/103 | <0.001 |
| Trehalose, µg/mg fly | Control | 13.44 | 0.61 | 4.27 | 0.27 | "Sex" | 569.5 | 1/72 | <0.001 |
| | Long-lived | 14.25 | 0.45 | 9.44 | 0.32 | "Diet" | 84.7 | 1/72 | <0.001 |
| | Salt | 14.72 | 0.38 | 7.03 | 0.30 | "Strain" | 36.6 | 3/72 | <0.001 |
| | Starch | 14.75 | 0.32 | 7.13 | 0.29 | Interaction | 11.5 | 3/72 | <0.001 |
| Trehalose-glucose ratio | Control | 2.96 | 0.34 | 0.76 | 0.15 | "Sex" | 269.8 | 1/72 | <0.001 |
| | Long-lived | 3.93 | 0.25 | 1.91 | 0.18 | "Diet" | 6.8 | 1/72 | <0.05 |
| | Salt | 5.27 | 0.21 | 1.37 | 0.16 | "Strain" | 18.6 | 3/72 | <0.001 |
| | Starch | 3.63 | 0.18 | 1.25 | 0.16 | Interaction | 13.6 | 3/72 | <0.001 |

Pairwise comparisons

| | Strain | Fly weight Mean | Fly weight SEM | Control | Long-lived | Salt | Starch | Glucose, µg/mg fly Mean | SEM |
|---|---|---|---|---|---|---|---|---|---|
| Fly weight, mg and Glucose, µg/mg fly | Control | 1.142 | 0.025 | | 0.56 * | 0.87 ** | 0.27 | 5.32 | 0.15 |
| | Long-lived | 0.946 | 0.025 | 0.144 *** | | 0.32 | −0.28 | 4.55 | 0.15 |
| | Salt | 0.937 | 0.024 | 0.149 *** | 0.006 | | −0.60 ** | 4.56 | 0.12 |
| | Starch | 1.024 | 0.024 | 0.145 *** | 0.001 | 0.005 | | 5.26 | 0.12 |
| | | | | Fly weight, mg | | | | Glucose | |

| | Strain | Trehalose Mean | Trehalose SEM | Control | Long-lived | Salt | Starch | Trehalose-glucose ratio Mean | SEM |
|---|---|---|---|---|---|---|---|---|---|
| Trehalose, µg/mg fly and Trehalose–Glucose ratio | Control | 7.33 | 0.27 | | −1.26 *** | −1.68 *** | −1.22 *** | 1.49 | 0.15 |
| | Long-lived | 11.84 | 0.27 | −4.85 *** | | −0.42 | 0.04 | 2.92 | 0.15 |
| | Salt | 9.59 | 0.23 | −3.92 *** | 0.93 * | | 0.45 | 2.67 | 0.13 |
| | Starch | 10.94 | 0.22 | −4.47 *** | 0.38 | −0.55 | | 2.44 | 0.12 |
| | | | | Trehalose, µg/mg fly | | | | Ratio | |

Notes: Mean and SEM: Sex-averaged (upper left part) and strain-averaged (lower part) values for fly weight, two main sugars in the fly's hemolymph, and their ratio. Upper part: Results of three-way ANOVAs. The independent factors: "Sex" (Male or Female), "Diet" (either standard food substrate for any strain or the same substrate as used in the process of selection of a strain); Interaction: "Sex" by "Strain" interaction. F: F-ratio; df: Degree of freedom; *p*: Level of significance. Lower part: the results of post hoc pairwise Bonferroni comparisons with * *p* < 0.05, ** *p* < 0.01, and *** *p* < 0.001 for t placed behind the estimate of difference between strains in strain-averaged mean.

Sleep was defined as behavioral quiescence and indexed as the absence of any locomotor activity for five or more consecutive minutes. See also other notes in the legend to Figure 1 and see Tables 1 and 2 for the results of rANOVAs of data illustrated in Figure 2a,b.

Finally, the comparison with the control strain suggested that rearing flies from the salt and starch strains on adverse food substrates also resulted in a significant reduction in the number of offspring (Table S2).

## 3. Discussion

Animal models can provide new insights into the evolutionary relationships between locomotor activity and various morphophysiological traits, development, health, and longevity. Here, we used four replicate strains of *Drosophila melanogaster* to test the following hypotheses:

(1) The evolution of flies toward a longer life and life on two adverse food substrates can cause an increase in locomotor activity and a decrease in sleep duration;

(2) Alternatively, these flies become less active and sleepier.

We found that flies selected to live either longer or on two adverse food substrates demonstrated a significantly enhanced locomotor activity and a significantly decreased sleep duration. In the case of such difference between the long-lived and control strain of *Drosophila melanogaster*, our results suggested that artificial selection for a long lifespan via delayed reproduction also selects for increased locomotor activity and decreased sleep duration. This association resembles the well-known difference between humans and ape species in terms of longevity and physical activity or sleep duration [24–26,84].

Notably, an enhancement of locomotor activity and, consequently, a reduction in sleep duration were consistently observed under rather diverse selection pressures. In particular, such similar responses were found to be caused by selection for two different durations: for imposing either earlier or later reproduction in flies from either the starch or long-lived strain, respectively. A possible reason for the similarity of responses to distinct selection pressures remains to be explored.

Our data on these responses of locomotor activity and sleep to various selection pressures are mostly consistent with previously published observations. For instance, the literature suggests that long-life-selected flies have an increased level of locomotor activity (e.g., [85]) and shorter sleep duration (e.g., [86]). Moreover, both short- and long-lived flies can display significantly higher locomotor activity than control flies [85]. However, the interpretation of these selection-induced changes in locomotor activity and sleep are not straightforward because the literature also indicates that different stress selections can cause a decrease rather than an increase in locomotor activity (e.g., [87]) and an increase rather than a decrease in sleep duration ([88,89]). Therefore, further research on experimental selection is required to clarify the effects of diverse stressing factors on the evolution of locomotor activity and sleep.

Since genes affecting locomotion are also likely to be involved in neurogenesis, metabolism, development, general cellular processes, etc. [42], we tried to answer question such as: can the selection-induced increase in a trait value (i.e., an increase in value in the case of locomotor activity) correlate with the responses of some of life-history traits? Namely, we expected that selection for late reproduction and selection imposed by rearing flies on two adverse food substrates can cause correlated changes in important phenotypic components of fitness such as body weight, the concentration of main insect sugars, longevity, and fecundity. Our results support the assumption that, under selection for either early or late reproduction, these traits can co-evolve with the traits of the 24 h patterns of locomotor activity and sleep.

Particularly, we found that, in each of the selected strains, the selection-induced reduction in fly weight was combined with an increase in their locomotor activity and a decrease in their sleep duration. Namely, our results suggest that flies from these selected strains, especially females, had lower body weight compared to the control flies. Such a response was expected in the case of selection on adverse food substrate and is in agreement with the responses reported in the previous studies (e.g., [90]). However, a similar response in the long-lived strain disagrees with the observations of May et al. [91], who indicated that selection for late-life reproduction extended lifespan and increased body weight.

On the other hand, it has to be noted that our result agrees with the previously reported findings of researchers who have conducted numerous evolutionary experiments on muroid rodents. They demonstrated a decrease in body weight in response to selection for increased voluntary wheel running. This relationship was explained by a negative genetic correlation between body weight and activity [92–94]. Moreover, such experiments showed that mice selectively bred for high voluntary wheel running can conserve more fat despite the increased exercise [95], and this spontaneous physical activity can negatively correlate with fat mass gain and obesity later in rodents' lives [96]. As for the association between physical activity and longevity in these and other mammal species, the comparisons of individual differences in body weight usually suggested that smaller individuals have higher rates of metabolism and live longer than their larger and slower conspecifics [97,98].

From the evolutionary perspective, selection can often favor the increase in the value of trait such as locomotor activity and the decrease in the value of such trait as sleep duration. Flies with these traits' values appear to have an evolutionary advantage over the unselected flies because they can increase the distance travelled outside of their home area for increasing the chances of encountering more potential sexual partners and food resources. A similar evolutionary advantage has been hypothesized for humans in their ancestral environment. More time spent awake and traveling has provided additional opportunities for a variety of productive activities, such as hunting, fishing, socializing, sex, etc. [26,99].

Our results on the effects of adverse food substrates on life-history traits seem to be in line with the previous publications showing that adaptation to a poor-quality diet generally selects for smaller fly weight and decreased fecundity [61,100–102]. Notably, flies from the starch strain had the lowest values of lifespan and fecundity among the studied strains, but they consistently demonstrated the highest value of locomotor activity. In natural settings, such an increase in this trait value in flies of this strain would allow, at least partly, the compensation of the disadvantages of other tested traits, such as a smaller body weight, a reduced lifespan, and a lower fecundity than in the unselected flies. Therefore, this result implies that enhanced locomotor activity and reduced sleep duration might be relevant to trade-offs among fitness-related traits (i.e., when an increase in one trait comes at the expense of other).

Studies in the laboratory have shown that mating and courtship rhythms are clock-controlled, with mating frequency being highest around the lights-on period [103–105]. The results obtained in laboratory settings were corroborated by data on the behavior of groups of flies in seminatural conditions. They showed that morning behaviors mostly comprise chasing, wing expansion, and copulation, which peak around dawn. Therefore, it was concluded that the morning peak of locomotion is mostly linked to courtship-associated activity [106]. Therefore, it is reasonable to suggest that the chance of copulation is higher in flies that are active earlier in the morning. The evening peak was related to general locomotion, to which no specific behavior can be assigned. Despite this, its predominant association with foraging-related behavior was proposed [106]. Therefore, enhanced and delayed evening activity can be beneficial for the purpose of encountering more food resources until the very end of the day. The characteristics of the morning and evening peaks of locomotor activity for flies from the selected strains can be regarded as advantageous for their survival and reproduction in the natural environment.

Interestingly, the adaptation of *Drosophila melanogaster* to a long photoperiod at higher latitudes appears to be limited by the inability of flies from this *Drosophila* species to delay a position of evening peak of activity relative to the position of the morning peak for more than 16 h interval [107–110]. Since the significant increase in lifespan imposed by selection for reproduction at an older age was accompanied by a profound change in the chronotype of long-lived flies; the "early to rise and late to bed" chronotype [110] can be a preadaptation to survival and reproduction at somewhat higher latitudes. As we expected, the 24 h patterns of the hybrids from the bidirectional breeding of the long-lived and starch flies demonstrated an intermediate 24 h pattern combining a high level of locomotor activity in one of the parent strains with a longer distance between the morning and evening peaks for another parent strain. Further research can also aim to uncover the genetic basis of the association of the evolutionary changes in longevity or diet with the changes in levels and 24 h patterns of locomotor activity and sleep.

Finally, Sujkowski et al. [111] reported that 65% of gene expression changes found in flies selectively bred for longevity were also found in flies subjected to three weeks of exercise training and that both selective breeding and endurance training increased endurance, cardiac performance, running speed, flying height, and levels of autophagy in adipose tissue and upregulated stress defense, folate metabolism, and lipase activity and downregulated carbohydrate metabolism and odorant receptor expression. These findings suggest the necessity for future studies to elaborate on the mechanisms underlying

the strong response of gene expression to selection for the extension of lifespan under an exercise-provoking environment.

Overall, the occurrence of correlated responses to selection may provide important clues for further investigations of the mechanisms responsible for the adaptation of complex traits to various selection pressures [40,112] Artificial selection by age at breeding and controlled natural selection with regard to the adverse food substrates can offer productive experimental approaches to understand the evolution of fitness-related traits. Our results illustrate the usefulness of such evolutionary experiments for testing the adaptive responses of activity, sleep, metabolism, reproduction, and longevity to distinct selection pressures. They particularly revealed that evolutionary changes in locomotor activity and sleep can, at least partly, compensate for the detrimental fitness effects on important life-history traits such as body weight, lifespan, and fecundity. Further studies might be essential for the validation of relevance of *Drosophila melanogaster* as a model for understanding the potential beneficial effects of the characteristics of the 24 h patterns of activity and sleep for the extension of the healthy lifespan and the success of adaptation to life in suboptimal environmental conditions.

## 4. Materials and Methods

### 4.1. Selection of Three Strains

In the evolutionary experiments, which started in September 2014, four strains of *Drosophila melanogaster,* called here "control", "long-lived", "salt", and "starch", were maintained [113–116]. All these strains originate from 30 wild flies caught in south-west Moscow in September 2014. The unselected and selected descendants of these flies were cultured at the Department of Biological Evolution of the Lomonosov Moscow State University (Moscow, Russia). Seven years later, in September 2021, flies from these four strains arrived from Moscow to Novosibirsk, where they were reared without further selection to use them in the experiments reported in this article.

After September 2014, the flies were housed in plexiglass boxes ($16.5 \times 16.5 \times 25$ cm$^3$). Each such box was supplemented with 12 open cylindrical glass tubes (10 cm height $\times$ 2.2 cm diameter) containing 10 mL of food. Each week, 4 of these 12 tubes were renewed. To reproduce flies, females and males were mixed (sex ratio ~ 1:1) to allow them to mate freely and lay eggs under a natural photoperiod and air temperature between 20 °C and 25 °C. Population density was not artificially regulated. The number of flies in a box varied from 200 to 700. Flies of the long-lived strain have been subjected to a sequence of selective breeding for the delayed reproduction, at the ages from 72 to 80 days [114]. Since the beginning of artificial selection in March 2018, as many as 15 selectively bred cohorts have been obtained (approximately four cohorts per year). Flies from this strain were called "long-lived" because they are expected to live longer than the control flies after several rounds of artificial selection imposed on the timing of reproduction. This expectation was confirmed in the present study (Table S1 and Figure S1). Since September 2014 and until recently, flies from the control and long-lived strains have been reproducing in the Moscow university department on the control food substrate (60 g of inactivated yeast, 35 g of semolina, 50 g of sugar, 45 g of crushed raisins, 8 g of agar, and 2 g of propionic acid per liter of food).

To maintain the salt and starch strains, the experiment on controlled natural selection started in January 2015. The selection was performed by placing flies either on "salted" food substrate (with additional 4% NaCl) or on starch-based food substrate (60 g of inactivated yeast, 30 g of starch, 8 g of agar, and 2 g of propionic acid per liter of food) [113,115,116]. For more than 6 years, flies from these two strains were forced to adapt to life on such adverse food substrates. The reproduction of these flies, especially the reproduction of flies from the starch strain, shifted at earlier ages due to the shortening of their lifespan due to selection (Table S1 and Figure S1).

The origin and maintenance of these four strains are described in detail elsewhere [113–116].

The samples of flies originating from the control, long-lived, salt, and starch strains were transferred to the *Drosophila* collection of the Institute of Cytology and Genetics (Novosibirsk, Russia) in September 2021. The same three Moscow food substrates were used for the further cultivation of flies from these 4 strains. Moreover, the strains were kept on two more food substrates, the standard substrate (18 g of dry yeast, 50 g of corn grits, 20 g of sugar, 40 g of raisins, 5.6 g of agar, and 7 mL of nipagin at 10%) and the low protein/high carbohydrate substrate (7 g of dry yeast, 50 g of sugar, 12 g of starch, 6.4 g of agar, and 4 mL of nipagin at 10%). Flies were fed standard and low protein/high carbohydrate diets in previous experimental studies of the 24 h patterns of locomotor activity and sleep conducted at the Department of Insect Genetics of the Institute of Cytology and Genetics (Novosibirsk, Russia) [110,117]. In order to further reproduce flies and use them in the experiments described in this article, they were kept in groups consisting of 10 males and 10 females in standard breeding vials (10 cm height × 2 cm diameter) supplemented with 10 mL of food per vial.

Finally, the same standard substrate was also used to rear the hybrids obtained by breeding flies from the long-lived and starch strains in both cross directions (i.e., by crossing ♂♂ from the long-lived strain with ♀♀ from the starch strain and ♀♀ from the long-lived strain with ♂♂ from the starch strain).

### 4.2. Testing the 24 h Patterns of Locomotor Activity and Sleep

Prior to the recording of locomotor activity and sleep, the same-sex groups of 20–25 flies from a strain were kept in standard vials under standard temperature (25 °C) and photoperiod (light between 7:00 and 19:00). We applied the conventional approach [118] to acquire and analyze locomotor activity using the DAMS (Drosophila Activity Monitoring System; "Trikinetics", Waltham, MA, USA) and the original software package (see the TriKinetics web site (Waltham, MA, USA): www.trikinetics.com, accessed on 12 February 2023). At the age of at least three days after eclosion, each fly was individually placed in a glass locomotor-monitoring tube of the DAMS with three sets of infrared beams for activity detection. To record beam breaks with one-minute intervals, the monitor was connected to a computer. Locomotor activity was recorded in constant darkness for at least 5 days under the same standard temperature (25 °C). The recorded locomotor activity was conventionally expressed as the numbers of beam breaks in 1 min bins. These data on locomotor activity were also used to quantify sleep events. They were defined, in accordance with Donelson et al.'s [119] criterion, as 5 consecutive minutes of absence of any locomotor activity. On the basis of this approach (www.trikinetics.com, accessed on 12 February 2023), our excel software was developed and used for the conventional analysis of locomotor activity and sleep (i.e., the summation of 1 min data on the 30 min intervals of each record prior to applying statistical analysis, Tables 1 and 2, and drawing illustrations of its results, Figures 1 and 2).

### 4.3. Testing Fly Weight, Content of Two Sugars, Longevity, and Fecundity

We also tested whether the difference in locomotor activity and sleep between four strains might be related to the differences in several important morphological and metabolic indicators of fitness (i.e., fly weight and concentrations of two main insect sugars), and whether the differences between strains in the life-history traits that are known to influence Darwinian fitness (i.e., survival rate and fecundity) persisted after the transferring of flies for their further cultivation from Moscow to Novosibirsk in September 2021. In Novosibirsk, the flies were left without further selection.

In the first of such tests, fly weights were measured in each of the 257 groups, including 10 (with further calculating the weight of one fly by the division of obtained weight of a group of 10 flies on 10). In the second of these tests, some of the weighted flies were randomly chosen to also measure the concentrations of two main insect sugars, glucose and trehalose, in the fly's hemolymph (Table 3). We used the assay that was mainly similar to the assay originally developed by Musselman et al. [120] and then slightly modified by

Karpova et al. [121]. Due to the difference between the strains in terms of fly weight, the concentrations of sugars are expressed as μg/mg fly (Table 3).

The methods and results of the tests on survival and fecundity are included in the Supplementary Materials (Figure S1 and Tables S1 and S2). The results of these tests are briefly described in both this Supplementary Materials and Results.

*4.4. Statistical Analysis*

The SPSS$_{23.0}$ statistical software package (IBM, Armonk, NY, USA) was used for all statistical analyses. To illustrate and analyze 24 h patterns of locomotor activity and sleep, the 30 min estimates obtained during the first day were excluded as mainly reflecting the habituation process. The estimates obtained for three consecutive days were averaged over each day to analyze and illustrate the 24 h pattern (i.e., the values of locomotor activity and sleep were calculated for 48 time points, constituting the 24 h cycle of a fly). Figures 1 and 2 illustrate the strain-averaged values obtained in two- and three-way repeated measure ANOVAs (rANOVAs; Tables 1 and 2, respectively). Degrees of freedom were corrected using Greenhouse–Geisser correction controlling for type 1 errors associated with the violation of the sphericity assumption, but the original degrees of freedom are reported in the upper parts of Tables 1 and 2. The significance of the difference between strains in the daily mean values of locomotor activity or sleep was examined using post hoc pairwise Bonferroni comparisons (Tables 1 and 2, lower parts).

To analyze the fly weight and concentrations of sugars, three-way ANOVAs were applied (Table 3, upper part). The effect of the factor "Strain" was further examined using a post hoc pairwise Bonferroni comparison (Table 3, lower part). See the Supplementary Materials for methods of testing, statistical analyses, and results on longevity and fecundity (Figure S1 and Tables S1 and S2).

**5. Conclusions**

*Drosophila* models of evolution under different selective pressures allowed use to examine whether increased locomotor activity is associated with the adaptation of this nonhuman species to a longer or harder life. We found that, when flies have been adapted to live longer and to live on adverse substrates, locomotor activity responds to different selection pressures in a similar way. Flies from the selected strains became more active and less sleepy than the control flies. Moreover, we found that selected flies can change their 24 h pattern of locomotor activity by extending the interval of enhanced locomotor activity at both earlier and later hours of the day. We speculated that such changes in locomotor activity might be relevant to trade-offs among fitness-related traits because a higher level of locomotor activity can confer an adaptive advantage and at least partly compensate for the detrimental effects of harsh selection pressure on several important fitness traits.

**Supplementary Materials:** The following supporting information can be downloaded at: https://www.mdpi.com/article/10.3390/clockssleep5010011/s1, "Supplementary to "Motus Vita Est: Fruit Flies Need to be More Active and Sleep Less to Adapt to Either a Longer or Harder Life"".

**Author Contributions:** Conceptualization, A.A.P., A.V.M. and L.P.Z.; methodology, A.A.P. and L.P.Z.; software, D.V.P.; validation, A.A.P. and L.P.Z.; formal analysis, A.A.P. and D.V.P.; investigation, L.P.Z., M.A.B., N.E.G., E.Y.Y. and A.V.M.; resources, L.P.Z. and D.V.P.; data curation, A.A.P.; writing—original draft preparation, A.A.P.; writing—review and editing, A.A.P., L.P.Z., D.V.P., M.A.B., N.E.G., E.Y.Y. and A.V.M.; visualization, A.A.P. and L.P.Z.; supervision, L.P.Z.; project administration, A.A.P.; funding acquisition, L.P.Z. All authors have read and agreed to the published version of the manuscript.

**Funding:** The study of E.Y. and A.M. was funded by Russian Science Foundation (#22-24-00227). The work of L.Z., D.P., M.B. and N.G. was supported by the Ministry of Science and Higher Education of the Russian Federation (project #FWNR-2022-0019).

**Institutional Review Board Statement:** Not applicable.

**Informed Consent Statement:** Not applicable.

**Data Availability Statement:** The dataset is available on reasonable request to the first author.

**Acknowledgments:** The study of E.Y. and A.M. was also associated with the scientific project under the state assignment of the Moscow State University (#121031600198-2). The North-Caucasus Federal University provided technical and other similar support to A.P. We are thankful to Ana K. Jones who devoted her time to editing this article.

**Conflicts of Interest:** The authors declare no conflict of interest.

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
