# Peer review of "Motus Vita Est: Fruit Flies Need to Be More Active and Sleep Less to Adapt to Either a Longer or Harder Life"

_2624-5175, doi:10.3390/clockssleep5010011_

Round 1

Reviewer 1 Report

I reviewed the Article with deep curiosity and passion. But it failed on many aspect in its current state. Firstly, It needs an English grammar correction thorrowly. Secondly, the results needs to be presented more ecological context including more studies on the animal and insect model. At present state, it seems like authors are forcefully trying to connect the results to the the human studies and requirement. Many other comments are marked on the manuscript file.    

Author Response

Reply to Comments of Reviewer #1

I reviewed the Article with deep curiosity and passion. But it failed on many aspect in its current state. Firstly, It needs an English grammar correction thorrowly.

Reply. To address this first of these “many aspect” (formulated as: “Firstly, It needs an English grammar correction thorrowly”), a native English speaker edited the revised version of the manuscript.

Secondly, the results needs to be presented more ecological context including more studies on the animal and insect model.

Reply. We addressed this second aspect (formulated as: “the results needs to be presented more ecological context including more studies on the animal and insect model”) by adding several new paragraphs in introduction and discussion sections of the revised manuscript. These paragraphs now include a number of citations of the results of various studies on laboratory evolution of most important fitness-relevant traits and related topics, these studies were mostly performed on insect and mammal models. Moreover, some of these cited publications reviewed the findings obtained in studies of animals from various taxonomic groups. Although our study is not related to any of ecological topics, some of our results and some results from the cited publications were discussed in relation to Drosophila ecology in the revised manuscript.

At present state, it seems like authors are forcefully trying to connect the results to the the human studies and requirement.

Reply. These mentioned above paragraphs that we added to Introduction and Discussion also address this comment (formulated as “it seems like authors are forcefully trying to connect the results to the the human studies and requirement”). In particular, they now include many new details that we think would be of importance for the understanding rationale of the experiments on these Drosophila models of evolutionary process. We stressed that these selection experiments that cannot be performed on humans provided insights into evolutionary relationships between activity, sleep, longevity, diet, and health in evolution of our species.

Many other comments are marked on the manuscript file.

Reply. Below, we tried to reply to these marked comments.

Reply to the comments in the entire text of the manuscript.

  1. Comment to the title: “adapt suits better rather evolve”

Reply. “evolve” was changed on “adapt” in the title.

  1. “physically” in introduction section of Abstract was excluded from the sentence: “Although higher physically activity takes more energy, humans have to remain physically active in and after middle age for increasing life span and reducing the risk of disease and death.

Reply. We replaced this sentence by another sentence that closer related to the following sentence (see 3).

  1. The sentence was excluded: “We used Drosophila model to examine whether locomotor activity increases and, consequently, sleep duration decreases due to evolution of a nonhuman species toward a longer or harder life.”

Reply. This sentence was rewritten to make its relation to the previous sentence clearer.

  1. Conclusion sentence of Abstract was excluded: “Fruit flies are becoming physically more active and sleep less when evolving to live longer or to live on adverse substrates.”

Reply. This sentence was rewritten, and the conclusion section of Abstract was extended to stress the benefits of the reported changes in the 24-h patterns of activity and sleep.

  1. Comment to the title “1.Introduction”: “Introduction is written more hypothetical way mostly connecting to humans rather presenting a proper ground for study using selective references published in model and non-model organisms.”

Reply. Introduction was profoundly extended to describe in more details the findings of the studies of animal models that can be of importance for the understanding rationale of the experiments on these Drosophila models of evolutionary process. They can provide new insights into evolutionary relationships between activity, sleep, longevity, diet, and health in evolution of our and other species. A number of new references was added to introduce the questions addressed in evolutionary experiments on laboratory animals. The emphasis was made on the description of two approaches, artificial selection and controlled natural selection, applied in the experimental Drosophila studies for testing adaptive responses of various complex traits including activity, sleep, stress resistance, body weight, metabolism, fecundity, and longevity.

  1. This was excluded: “Although higher physically activity takes more energy, humans had to be physically active during the whole lifespan. In particular, we never evolved to prevent senescence in the absence of physical activity. Instead, we evolved to need physical activity for the extension of health and life spans.”.

Reply. This is what the authors of cited papers said. We corrected these sentences but tried to convey the idea formulated by the authors of these papers. Moreover, we added two new paragraphs prior to this sentence for better understanding this idea and its relationship with this direction of evolutionary studies.

  1. Comment to Table 1“This table needs to be supplemented with p values. Although autheors listed the p-values in the table footnote but its reads very complex and hard to understand.”

Reply. As requested, we indicated these p-values (they all are ***: p <0.001) in the table cells and explained this in the table notes.

  1. Comment to the title of Table 1.: “Why some? seems vague. Be presice and clear with data writing.”

Reply. This “Some” was excluded from this and other tables.

  1. Comment in Table 1 to “3/873”: “make sure its df value. currently its listed wrongly.

Reply. We pretty sure that these degrees of freedom are correct since they were directly taken from the results of analysis in SPSS. If this is a question on the way of presenting the degrees of freedom, this is the standard way of reporting degrees of freedom in ANOVA (See, for instance, Walker, H. W. (1940). Degrees of Freedom. Journal of Educational Psychology, 31(4), 253–269). Namely, this “3/873” means “df1/df2 = 3/873”. Here, Df1 and df2 refer to two degrees of freedom. Df1 is about means and not about single observations. In this case, it is equal to 3 because there were 4 strains compared (df1=n-1). Whereas df1 is about how the cell means relate to the grand mean or marginal means, df2 is about how the single observations in the cells relate to the cell means. Therefore, Df2 is calculated as the total number of observations in all cells (n) minus the degrees of freedoms lost because the cell means are set (that is, minus the number of cell means, k). Thus, this df2 is calculated as: minus k plus the number of flies multiplied by the number of time points minus 1 (for this case, 4 time points – 1 = 3).

  1. Comment in Results: “what authors mean by also here?”

Reply. Previously, we described significant difference in mean levels of locomotor activity and sleep. In this paragraph we wanted to say that there was also significant difference in the wave-form of the 24-h curves supported by rANOVAs. In the revised version, we stated it clearer: “…three selected strains showed significant differences from the control strain not only in mean levels but also in the 24-h patterns of locomotor activity and sleep”.

  1. Comment to X-axis of Figure 1: “See if authors can present these time values in more standard format.”.

Reply. We checked the submitted docx and pdf files. Everything is ok with this axis in our files. Our guess is that someone from the journal office inaccurately re-arranged the submitted files (already after their submission). Hope the revised docx and pdf files will not be treated as badly as the previously submitted files.

  1. Comment to the title “Figure 1”: Its quite strange that authors choose to present 30 and 360 minute interval data differently (line vs bar plot). Is there any logic for this kind of disparity. Further the selection of these 2 time points are not justified. Authors should justify the selection of these 2 time points and its possible advantages.

Reply. Practically everyone who study Drosophila’s locomotor activity and sleep uses this Drosophila Activity Monitoring System ("Trikinetics", Waltham, Massachusetts, USA), with the software package posted at the TriKinetics web site (www.trikinetics.com). Therefore, all researchers use the 30-min intervals for the analysis and illustrations. Since there are, we guess, thousands of such publications, it need not justify the choice of this 30-min interval. Therefore, we enlarged this section of Methods to stress that we collected and analyzed data in the conventional way. Moreover, we added one more reference. The cited paper contains the detailed description of this method. About the graph with a longer (360-min) interval. Since analysis on such longer intervals is optional and, in our study, its results did not add important information to the results on the 30-min intervals, we simply excluded it from Tables 1 and 2 and Figures 1 and 2. Instead we included the results on more recently collected data, on the hybrids of two strains (Table 2, right, and Figures 1BC and 2BC).

  1. Comment to the title “3.Discussion”: “needs to be re-written based on results and supported by proper citations. Currently, I feel its very basic and lacks deep connections to the results.”

Reply. After providing much more details in Introduction, we also have rewritten Discussion to describe our results in more details, i.e., to compare them with a number of previously published reports (mostly with the results of similar evolutionary experiments), to provide the interpretation of some of our results in the light of the existing literature, to wider discuss the results obtained on life-history traits of Drosophila, etc.

  1. Comment to the section “4.1. Selection of Three Strains from Control Strain from Wild Population”: “Not clear enough, re-write!”

Reply. We did. This section was extended by adding two paragraphs with detailed description of two evolutionary experiments preceding the experiments described in our manuscript.

  1. Comment to the title “5.Conclusions”: “Conclusion needs to be connected for the larger hypothesis and goals in subjects. Bringing more ecological aspect can solve this issue.”

Reply. This section was almost fully rewritten and extended in the light of new texts added to the introduction and discussion sections. These paragraphs now include resume on the results on laboratory evolution of various fitness-relevant traits with mentioning some of related topics. Although our study is not related to ecology, we included sentences on a related topic of adaptive advantage of higher locomotor activity and lower sleep duration, (e.g., that this advantage can, at least, partly compensate the reductions of some of important fitness traits, especially under a harsh selection pressure).

  1. Comment to the title “References”: “See if authors can improve the reference list , only 10 out of 26 are from last 5 years. Furthermore, many citations are not properly cited and incomplete (e.g., 3&4).

Reply. We do not find that the citations 3&4 were incomplete and/or not properly cited. Our guess is they might look as incomplete because these are the references of books, not articles. Nevertheless, we excluded all book references from the revised version of the manuscript. We also think, in many cases, we need not replace the references of priority research by the references of their more recent epigones. Nevertheless, we enriched the revised version of the manuscript by adding many last-5-year publications. Finally, we unified all references in the enlarged reference list.

Reviewer 2 Report

The manuscript entitled as "Motus vita est: fruit flies need more activity and less sleep to evolve for either longer or harder life" is very interesting article. However, I would like authors to address some points and incorporate them in the manuscript:

1. In materials and methods what is the criteria of selection of the strains used for experiments

2. How many females and males were used for this experiment

3. Mention the density of female and males kept separately in vials before experiment

4. How the selection experiment was performed

5. Whether authors took healthy flies for experiment and what is the body weight of flies before experiment

6. Discuss the relationship between weight and locomotion of flies which may impact sleep

Author Response

Reply to Comments of Reviewer #2

The manuscript entitled as "Motus vita est: fruit flies need more activity and less sleep to evolve for either longer or harder life" is very interesting article. However, I would like authors to address some points and incorporate them in the manuscript:

  1. In materials and methods what is the criteria of selection of the strains used for experiments

Reply. Section “4.1. Selection of Three Strains” was extended by adding two paragraphs with detailed description of these two evolutionary experiments.

  1. How many females and males were used for this experiment

Reply. The number of flies in the selection experiment was provided in the 1st of these paragraphs. The numbers of flies from each strain in the experiments described in this manuscript are given in the notes to Tables and legends to Figures. The total number of flies for which the recordings of locomotor activity were obtained is given in Abstract.

  1. Mention the density of female and males kept separately in vials before experiment

Reply. The numbers of female or male flies per each standard vial were added to the beginning of section “ 4.2. Testing the 24-h Patterns of Locomotor Activity and Sleep”, and the number of flies participated in reproducing flies of four strains is given in the end of the previous section.

  1. How the selection experiment was performed

Reply. See the reply to the 1st comment.

  1. Whether authors took healthy flies for experiment and what is the body weight of flies before experiment

Reply. Flies were weighted only prior to performing the special measurement of concentrations of two major insect sugars. Some the weighed flies were then randomly chosen for this measurement. We clarified this in the revised text of the section “4.3. Testing Fly Weight, Content of Two Sugars, Longevity, and Fecundity”.

  1. Discuss the relationship between weight and locomotion of flies which may impact sleep

Reply. We added two new paragraphs in Discussion in which we reviewed the results of our and previously published evolutionary experiments on laboratory flies and rodents on the relationship between weight and locomotion.

Round 2

Reviewer 1 Report

The manuscript underwent a significant amount of revision, which resulted in it reading more smoothly. There are still some questions I have regarding the manuscript (marked on file).

Author Response

Reply to Comments of Reviewer #1

The manuscript underwent a significant amount of revision, which resulted in it reading more smoothly. There are still some questions I have regarding the manuscript (marked on file).

Reply. We made all suggested in the attached text corrections. See below.

Reply. Below, reply to the suggestions made in the text.

Reply to the comments in the entire text of the manuscript.

  1. The sentence was excluded by the reviewer from Abstract: “Humans have evolved to become more physically active than apes. This.”

Reply. We agree with this correction. This sentence was excluded. The next sentence starts with “Activity…”

  1. The sentence was excluded by the reviewer from one of the paragraphs of Introduction (page 2, upper sentences, line 50): “It is required for localization of food and mates, defense of territory, escape from predators, response to various stresses, etc..”

Reply. As recommended, this sentence was excluded.
